# The weather affects air conditioner purchases to fill the energy efficiency gap

Pan He [1], Pengfei Liu [2] ✉, Yueming (Lucy) Qiu [3] ✉ & Lufan Liu[2]

Energy efficiency improvement is often hindered by the energy efficiency gap. This paper examines the effect of short-run temperature fluctuations on the Energy Star air conditioner purchases in the United States from 2006 to 2019 using transaction-level data. Results show that the probability of purchasing an Energy Star air conditioner increases as the weekly temperature before the transaction deviates from 20–22 °C. A larger response is related to fewer cooling degree days in the previous years, higher electricity prices/income/ educational levels/age/rate of owners, more common use of electricity, and stronger concern about climate change. 1 °C increase and decrease from 21 °C would lead to a reduction of total energy expenditure by 35.46 and 17.73 million dollars nationwide (0.13% and 0.06% of the annual total energy expenditure on air conditioning), respectively. Our findings have important policy implications for demand-end interventions to incorporate the potential impact of the ambient physical environment.

Energy efficiency improvement is critical to Net Zero emissions. Global energy consumption has been increasing ~1.85% annually during the past four decades with the total demand more than doubled[1]. Boosting the adoption of energy-efficient technologies is recognized as an essential part of realizing the Net Zero goal by 2050, which also brings other benefits such as reduced energy cost and air pollution[2]. However, such changes are often hindered by the underinvestment in effective energy-saving techniques and products known as the energy efficiency gap[3,4]. Behavioral anomaly is one of the possible causes of the observed energy efficiency gap[3]. Consumers are required to predict future utility when making investment decisions such as purchasing appliances as durable goods according to the standard economic framework. In addition, such predictions need to be accurate and consistent regardless of time and place, which has been challenged by recent evidence from behavioral economics where consumers could heavily discount fuel economy in the future[5,6], and conflicting evidence on whether consumers pay limited attention to the attribute of energy efficiency in their purchasing decisions[7,8]. As a result, the benefit of energy-saving products is underestimated and these products are thus less consumed.

The change in the physical environment can also impact the purchase decision by reshaping the predicted future utility of consumers. Recent weather has been demonstrated to affect various purchasing decisions, including types of automobile purchases[9], solar panel adoption[10], catalog order of clothing[11], college enrollment[12], etc. Poor air quality increases the near-term health insurance purchase[13], the transaction price of housing purchase[14], and willingness-to-pay for pollution control[15], while the salience of flood risk would raise the purchase of flood insurance[16]. A rationale behind these linkages lies in projection bias with which consumers anticipate the future ambient environmental conditions to resemble the current status, despite what they are experiencing is a short-run fluctuation of the conditions rather than a long-term change[9,17]. Another explanation is salience indicating that the attention of consumers is directed to the environmental features that are more valued in making purchasing decisions[18,19]. It is thus reasonable to expect the fluctuation of the physical environment will affect the choice of consumers via similar mechanisms related to the energy efficiency gap. However, it is unclear whether the short-term change in environment, or more specifically, temperature, could help overcome the energy efficiency gap, which is underexplored in the current literature.

[1]School of Earth and Environmental Sciences, Cardiff University, Cardiff, UK. [2]Department of Environmental and Natural Resource Economics, College of the Environment and Life Sciences, University of Rhode Island, Kingston, RI 02881, USA. [3]School of Public Policy, University of Maryland College Park, College Park, MD 20742, USA. ✉e-mail: pengfei_liu@uri.edu; yqiu16@umd.edu

This paper examines the effect of short-run weather fluctuations on air conditioner purchases in the United States. Using Nielsen Retail Scanner Dataset, we access individual purchase records of air conditioners in the United States from 2006 to 2019. The purchase records are summarized by the model of the air conditioner and week of purchase and then matched with the ground station climate records aggregated from the dataset of Global Surface Summary of the Day[20]. We conduct transaction-level regression analysis to examine whether the temperature deviating from the thermal comfort in the week ahead of the purchase would increase the probability of purchasing an Energy Star-certified model. The Energy Star program was initiated by the Environmental Protection Agency in 1992 aiming at promoting energy-efficient appliances by awarding certain types of products and devices a label of Energy Star if their energy efficiency is above a specific threshold. Results show that a weekly average temperature either higher or lower than 20–22 °C will increase the probability of purchasing an Energy Star model versus a non-efficient one. Such effect is realized through both projection bias and salience pathways. Further investigation of heterogeneous effect finds that the stimulation effect of temperature would be amplified by higher state-level electricity prices, fewer historical Cooling Degree Days (CDDs), higher household income, higher educational levels, more intensive concern about climate change, and higher support for the Democratic Party.

This study contributes to the energy and behavioral economics literature by adding empirical evidence on the positive effect of projection bias and salience for environmental benefit. Our results indicate a possible way of narrowing the energy efficiency gap with weather-related nudging strategies, which can be further fueled by stressing the critical impact of climate change and the consequential fluctuation of temperature in the future. Given the considerable global heat exposure, especially in the emerging economies which can be further exaggerated by the future climate change[21], our findings have profound implications for not only policymakers but also enterprises in designing effective interventions to stimulate demand-side changes to incorporate the potential impact from the ambient physical environment for enlarged effectiveness.

## Results

### Effects of weather on Energy Star air conditioner purchase

We first conduct a generalized linear regression model to investigate whether the temperature and other meteorological indicators affect the Energy Star air conditioners purchases in the coming week. The results show that deviation of weekly average temperature from comfort before the transaction encourages the purchase of Energy Star relative to the non-Energy Star air conditioners. The probability of purchasing an Energy Star air conditioner significantly increases as the ambient temperature rises from 20 to 22 °C (Fig. 1a and Column 1, Supplementary Table 2). This interval of thermal comfort approximates but is slightly higher than the most favored temperature of 65 °F (about 18 °C) in the United States[22]. The probability of purchasing an Energy Star air conditioner increases by more than 2% when the temperature shift raises to 24–32 °C, and by more than 5% when the temperature exceeds 32 °C. By contrast, colder weather encourages such transactions by 0.7–3%. Although the air conditioner is mainly adopted for cooling, some models are equipped with the heat mode and thus can be used for heating, explaining the increase of Energy Star products as the temperature drops from 20 to 22 °C. As the temperature further deviates, consumers may shift to other heating appliances such as heat pumps, electric heaters, etc., and thus the Energy Star air conditioners become less competitive.

We also use the temperature deviation from 21 °C (positive and negative) as regressors to estimate the marginal effects. The magnitude of changes is similar for turning hot and cold as the coefficients of positive and negative deviation are similar in magnitude (Column 3, Supplementary Table 2), indicating that 1 °C increase from 21 °C will raise the probability of purchasing an Energy Star conditioner by 0.4% while 1 °C decrease will lift the probability by 0.2%.

We adopt weekly accumulative cooling degree days (CDDs) and heating degree days (HDDs) with 70 °F (~21.11 °C) as the reference point for robustness checks as well (to be consistent with the reference temperature interval of 20–22 °C, Column 1, Supplementary Table 3). In addition, since some transaction prices are much smaller than the common market price of an air conditioner ($150–$750), we dropped

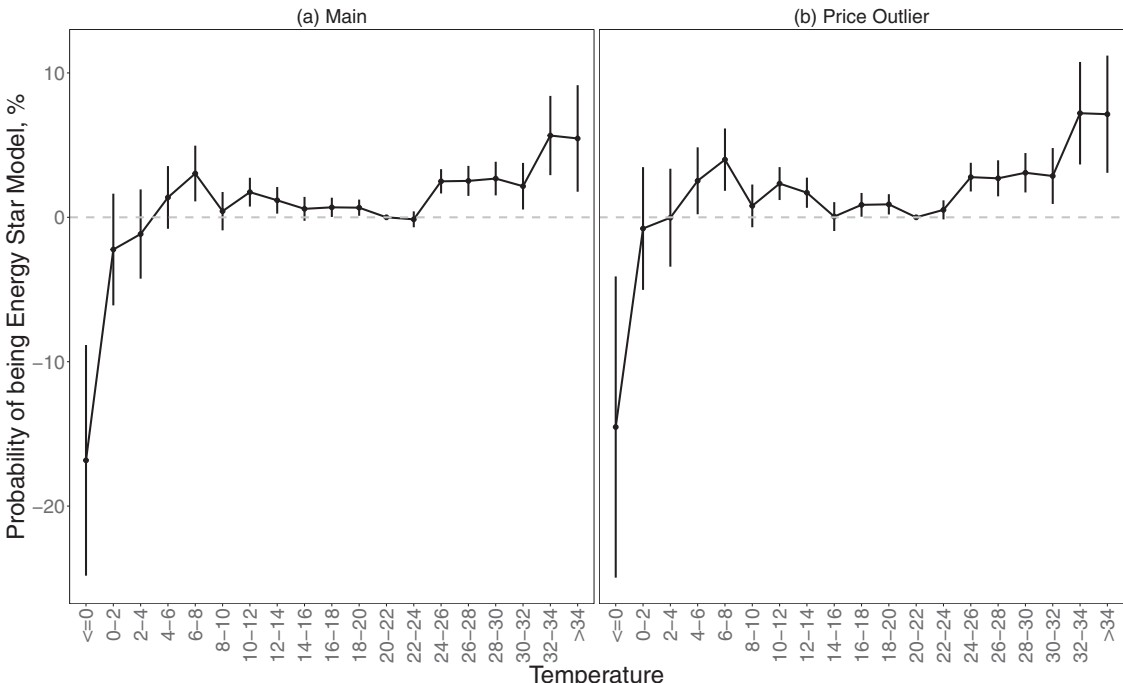

**Fig. 1 | The effect of temperature on Energy Star air conditioner purchase. a** Full sample (N = 1,871,472). **b** Records with price < $100 dropped (N = 1,564,506). All the sample sizes refer to transactions from the Nielsen Scanner Dataset. The statistics come from the two-sided t-tests based on the regression analysis with the full results shown in Supplementary Table 2. The vertical lines show the 95% confidence intervals (i.e. mean values ± 2SEM).

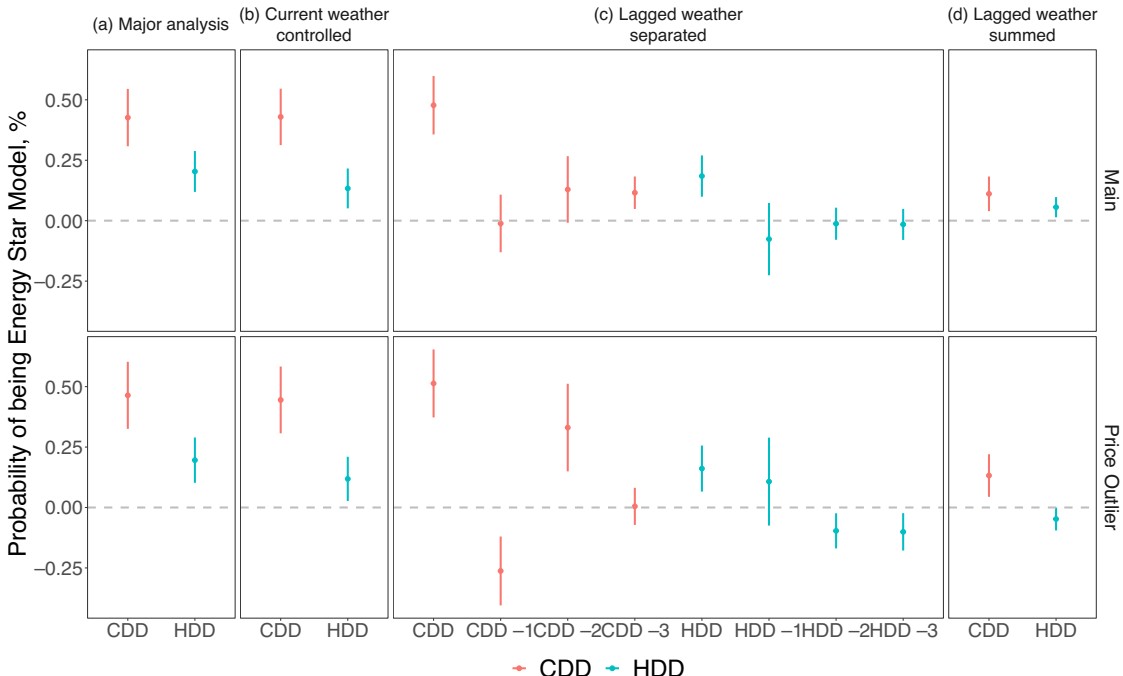

**Fig. 2 | The effect of temperature on Energy Star air conditioner purchase in terms of cooling degree days (CDDs) and heating degree days (HDDs). a** major analysis using the CDDs and HDDs in the previous week of transaction ($N = 1,871,472$ for the full sample, 1,564,506 for the records with price < $100 dropped), **b** controlling for the meteorological records in the week of transaction ($N = 1,868,326$ for the full sample, 1,561,994 for the records with price < $100 dropped), **c** controlling for the meteorological records lagged for 1, 2, and 3 weeks ($N = 1,862,247$ for the full sample, 1,557,351 for the records with price < $100 dropped), **d** impact of the summed CDDs and HDDs in the previous month of the transaction ($N = 1,862,247$ for the full sample, 1,557,351 for the records with price < $100 dropped). All the sample sizes refer to transactions from the Nielsen Scanner Dataset. The statistics come from the two-sided $t$-tests based on the regression analysis with the full results shown in Supplementary Table 3. The vertical lines show the 95% confidence intervals (i.e. mean values ± 2SEM).

the records with a transaction price of <$100 (no records with abnormally large prices are dropped as the maximum price in the dataset is $699.99 as shown in the descriptive statistics in Supplementary Table 1) and rerun the regression. Our results remain robust with magnitudes increased slightly as shown in Fig. 1b and Columns 1 and 2, Supplementary Table 3.

**Robustness checks**

We conduct several robustness checks to increase the credibility of our results. As the weather can be temporally autocorrelated, the temperature in the previous week can be correlated with the temperature in the week of the transaction, which can impact the outdoor activity of individuals. If the geographical distribution of Energy Star differs from that of non-Energy Star air conditioners, e.g., an individual would need to travel further to obtain an Energy Star product, such autocorrelation may threaten the validity of our results. We thus control for the weather indicators of the current week (Fig. 2b, and Columns 3 and 4 in Supplementary Table 3), and our results remain robust.

Another alternative explanation is the harvesting issue if the consumers who are more prone to Energy Star products bring forward their purchase decisions because of the fluctuation of temperature, we may still obtain a positive correlation between the CDDs/HDDs and Energy Star air conditioners. However, this should not affect the changes in the types of purchases. To exclude such a possibility, we conduct two additional tests following previous studies[9,23]: (1) adding the weather indicators in 1, 2, and 3 weeks before the period of interest referring to a previous study (Fig. 2c, Columns 5 and 6 in Supplementary Table 3), and (2) aggregating the weather indicators in the 4 weeks covered in (1), i.e. in the month before a transaction takes place (Fig. 2c, Column 5 and 6 in Supplementary Table 3). We find the coefficients associated with CDDs and HDDs remain significant after controlling for the weather of lagged weeks. The magnitudes of the

monthly level weather coefficients decrease but are still statistically significant, indicating that at least part of the effects can be attributed to changes in types of purchase instead of temporal reallocation of decision, despite potential harvesting behaviors.

The relationship between temperature and Energy Star purchase probability can be attributed to two possible mechanisms: projection bias and salience. Projection bias indicates that the consumer may amplify how the future weather resembles the present experience when predicting their utility and thus assign a higher likelihood of hot/cold days[17,24,25]. Salience implies that customers' attention can be systematically directed toward cost-saving of energy-efficient appliances and become more salient on hot/cold days[19]. Although both mechanisms imply an increase in energy-saving product purchase with temperature deviation, the increase is due to projection bias when the weather is hot/cold in absolute values while due to salience when the weather is hotter/colder than a benchmark defined using recent weather[9]. In this sense, the consumers are prone to the Energy Star products either on a hotter or colder day at least partially because of salience as the coefficients of CDDs and HDDs remain stable. Provided that the Energy Star air conditioners are of higher prices on average (Supplementary Table 1), the two mechanisms are likely to enlarge the value of consumers on the operational cost rather than the capital cost of durable goods. To date, evidence is mixed on whether the consumers are forward-looking or myopic in durable good purchases[7,8], yet our findings suggest that short-run weather changes may add to the rationality of consumers.

As the reference temperature adopted in the analysis (20–22 °C for intervals and 70 °F for CDDs and HDDs) is slightly higher than what is used in the previous research[22] (16–18 °C for intervals and 65 °F for CDDs and HDDs), we switch to these reference interval/threshold to test whether our key findings still hold. The results shown in Supplementary Table 4 indicate similar statistical significance and magnitudes.

Since temperature deviation as well as meteorological indicators other than the relative humidity may also have a non-linear effect, we introduce the quadratic terms of each weather indicator and rerun the regression. As shown in Supplementary Table 5, the effect of temperature deviation from 21 °C by and large stay linear as in the main analysis, although the probability of Energy Star transactions first increases and then decreases when the temperature drops in the full sample, aligning with Fig. 1 that indicates a potential shift to other heating measures when the temperature becomes lower than around 9.5 °C. The introduction of the quadratic terms of other meteorological indicators does not change these results.

In case our results are sensitive to the setting of standard error clustering (store level), we test different clustering levels available in the dataset including county, the first 3 digits of the zip code zone, and the Designated Market Areas (defined by Nielsen to divide the country into more than 200 regions in which consumers are faced with same options of major media and thus the similar information flows from advertisement). The significance of CDDs and HDDs remain with a *p*-value smaller than 0.001 in all the tests (Supplementary Table 6).

In case our results are sensitive to the setting of fixed effects, we switch to alternative combinations of fixed effects and rerun the major analysis on CDDs and HDDs. As shown in Supplementary Table 7, the coefficients of both terms remain significant with a similar magnitude, indicating the robustness of our results.

It is possible that the weather affects the sales of air conditioners via the change of price, e.g. promoting Energy Star air conditioners with larger discounts compared with non-Energy Star products as CDDs/HDDs increase. We test this possibility by interacting the dummy of Energy Star in the transaction with the meteorological variables and run a regression of these variables on the transaction price (in log form). As shown in Supplementary Table 8, the Energy Star products are of a lower price when the weather is either hotter or colder in the full sample, but the significance turns to the opposite sign for CDDs after the exclusion of the price outliers. Although it is hard to reach a meaningful conclusion based on the regression results, we have controlled for the price with a weekly variation in the main regression so that the effect of price fluctuation on the purchase decision is controlled and would not threaten our key findings.

To testify whether the statistical significance only indicates a spurious correlation, we conduct a falsification test replacing the air conditioners with telephones. Short-run weather should not impact the sales of Energy Star products that are not sensitive to weather changes. As the usage of a telephone is hardly affected by temperature, the coefficients of temperature intervals/CDDs/HDDs should remain insignificant. The results of the test are presented in Supplementary Fig. 2, Columns 5, and 6 in Supplementary Table 2, as well as Supplementary Table 9, where the indicators of temperature are both rarely statistically significant and of small magnitude, indicating that our estimation of air conditioners is less likely to be affected by unobserved confounders.

### Heterogeneous effects of temperature
The effect of weather can vary given the individuals' heterogeneous preferences when predicting the future utility. Energy Star products can be further merited where the cooling and heating are more costly due to larger energy usage or a higher electricity price. We explore the effect of average county-level annual CDDs and HDDs, as well as the mean state-level electricity price (all during the study period of 2006–2019), on the Energy Star air conditioner purchase decisions. We matched the weather and price data with the air conditioner purchase sample and examine the coefficients of CDDs/HDDs by running regressions on tertile groups. Results show that individuals are more affected in states with higher average electricity prices (Fig. 3a, and Column 1, Supplementary Table 10). They also tend to have a stronger response with fewer background CDDs in the previous years (Fig. 3b,

and Column 2, Supplementary Table 10). The HDDs, however, show a non-monotonic effect on the extent of response. The response to temperature first rises and then declines as the background HDDs increase (Fig. 3c, and Column 3, Supplementary Table 10). This may imply the existence of salience as individuals are more sensitive to temperature change when the background temperature is milder so that the importance of energy saving starts to emerge. Another possible reason is that in a more drastic climate, consumers tend to resort to other cooling/heating appliances rather than window air conditioners such as central cooling/heating systems, heat pump, etc.

Socio-economic characteristics may also influence the Energy Star product purchase response to temperature changes. Since the consumer characteristics are not available at the transaction level, we use the county-level median income, the proportion of the population over 25 with an education degree higher or equal to a bachelor's, and the proportion of White people from the 5-Year American Community Survey from the U.S. Census as a proxy portrait of consumers. We find that while the response tends to increase with income and educational attainment (Fig. 3d–g), the difference is not statistically significant for the latter (Column 2, Supplementary Table 11). The proportion of White people does not affect the level of response (Column 3, Supplementary Table 11). Higher age is related to a larger response (Column 3, Supplementary Table 11), possibly because elder individuals are more sensitive to the temperature or the positive correlation between age and income.

Housing characteristics may differ in the response as well. We examined the effect of the median number of rooms in housing units, the ratio of owners to renters, and the percentage of electricity as the heating fuel (Fig. 3h–j). We find that the rooms of housing units do not significantly affect the level of response (Column 1, Supplementary Table 12), despite that the central heating system may benefit more housing units with more rooms and discourage the purchase of window air conditioners. Counties with more owners have a larger response to CDDs but a slightly smaller response to HDDs (Column 2, Supplementary Table 12), likely implying the presence of principal-agent issues[3]. A higher proportion of housing units using electricity as heating fuel leads to a smaller response (Column 3, Supplementary Table 12), indicating the adoption of central temperature control systems.

We also investigated how attitudes towards climate change could make a difference. The metrics are retrieved from a previous study[26] in terms of scores ranging from 0 to 100 reflecting what extent individuals in each county believe climate change happening, worrying about climate change, as well as supporting regulation of $CO_2$ as a pollutant in 2013. In addition, we also obtain the proportion of support for the Democratic Parity in the county presidential election during 2000–2020[27]. A stronger attitude towards actions of climate change is shown to be related to a larger response (Fig. 3k–p), with the difference being statistically significant for the belief of climate change harming the US, worrying about climate change, supporting regulation of $CO_2$, and supporting the Democratic Party (Supplementary Table 13).

## Discussion
Our results demonstrate that deviating from the comfort temperature range significantly increases Energy Star product purchases relative to non-efficient ones. We show that significant adoption responding to climate changes exists since consumers alter energy-efficient appliance purchases when the temperature fluctuates. Such adoption may mitigate the negative impact of climate change as energy-efficient appliances contribute less to carbon emissions. Provided that the annual average air conditioner shipped in the US during 2006–2019 is approximate 4.44 million[28], our estimations imply that Energy Star conditioner purchases will increase by 17,760 and 8880 with 1 °C increase and decrease from 21 °C, respectively, as a replacement of current transactions. An Energy Star air conditioner could use 10% less

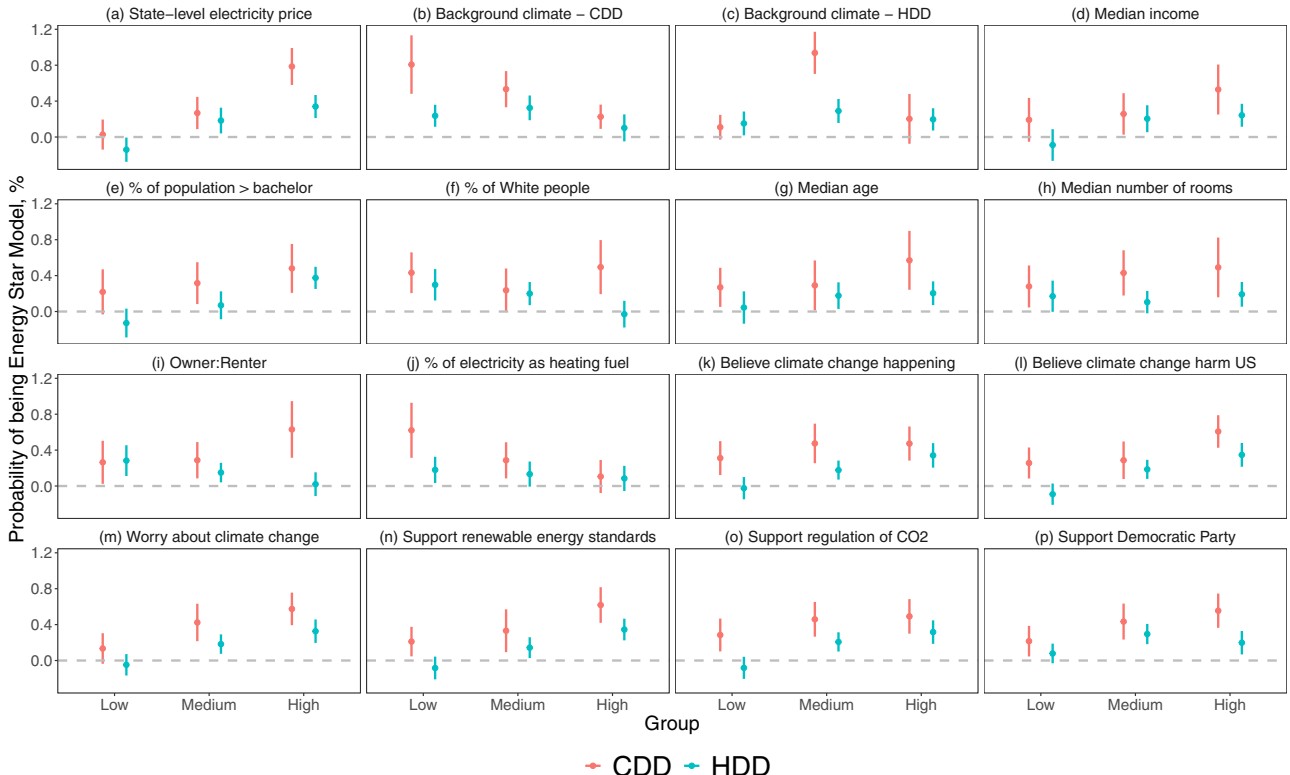

**Fig. 3 | Heterogeneous effect of temperature in terms of cooling degree days (CDDs) and heating degree days (HDDs). a** State-level electricity price ($N$ = 1,871,472). **b** CDDs in the decade before the study period ($N$ = 1,871,326). **c** HDDs in the decade before the study period ($N$ = 1,871,326). **d** Median income level in the county ($N$ = 1,474,685). **e** percentage of population with an educational level higher than bachelor in the county ($N$ = 1,474,685). **f** Percentage of White people in the county ($N$ = 1,474,685). **g** Median age in the county ($N$ = 1,474,685). **h** Median number of rooms of the housing units in the county ($N$ = 1,474,685). **i** The ratio of owner and renter of the housing units in the county ($N$ = 1,474,685). **j** Percentage of electricity as the heating fuel of the housing units in the county ($N$ = 1474685). **k** Scores of the county in believing that climate change is happening ($N$ = 1,870,598). **l** Scores of the county in believing that climate change harms the US ($N$ = 1,870,598). **m** Scores of the county in worrying about climate change ($N$ = 1,870,598). **n** Scores of the county in supporting renewable energy standards ($N$ = 1,870,598). **o** Scores of the county in supporting regulation of $CO_2$ ($N$ = 1870598). **p** Scores of the county in supporting democratic party ($N$ = 1,869,917). The vertical lines show the 95% confidence intervals (i.e. mean values ± 2SEM). All the sample sizes refer to transactions from the Nielsen Scanner Dataset. The statistics come from the two-sided $t$-tests based on the regression analysis with the full results shown in Supplementary Tables 10–13. The groups are divided using tertiles of the variable that indicates potential heterogeneous preference.

energy and help to save an averagely $75 per unit of air conditioner[29] accounting for ~28.3% of the home energy expenditures on air conditioning in the United States in 2015[30] which could possibly be even higher provided the trend of future climate change. In this way, 1 °C increase and decrease from 21 °C would help to save electricity use by 292.4 and 146.2 billion Btu (60,730 and 30,365 tons of $CO_2e$, provided that the air conditioning accounts for 731 trillion Btu of the household energy consumption in 2015[30]) compared with the status quo of energy consumption from air conditioners in a counterfactual situation of constant climate, respectively, as a back-of-envelope estimation. Despite that the magnitude is not as large as the production-side mitigation techniques, such effects enlighten the opportunities for low-cost nudging strategies to enlarge the perception of and sensitivity to weather in consumer decisions. It would also lead to a reduction of total energy expenditure by 35.46 and 17.73 million dollars (0.13% and 0.06% of the annual total energy expenditure on air conditioning) at the national level compared to the scenario when we assume the temperature would not impact the probability of purchasing an energy star certified air conditioner, respectively. This benefit may be partially canceled out if the temperature change also encourages new purchases of Energy Star products among those who would have lived without air conditioning in a constant climate. Quantitative estimation of the magnitude of this effect is not available given the limitation of our data. Nevertheless, it could be small provided that the penetration rate of air conditioning has hit 88% in 2020[30].

Given the considerable reduction of energy bills, making use of the effect of weather to encourage the purchase of an energy-efficient air conditioner would help relieve energy poverty and further reduce the adverse health impact due to extreme heat by adding to the affordability of energy among the poor population. Our tests of heterogeneous effects show that consumers demonstrate a higher possibility of a shift to Energy Star products when they are faced with higher electricity prices or fewer cooling degree days in the previous years; similar is the case when they have higher income, higher educational levels, or stronger concern of the climate change issues and support to responsive actions. In this way, nudging strategies combining the effect of weather should be developed in adaption to the characteristics of this population, while more research is required on how the population of low-socio-economic status could be further encouraged.

Several limitations should be noted. This study is conducted based on weekly transactions with the meteorological records aggregated and thus may lose daily fluctuation of weather that impacts the decision of purchase. Moreover, although the Nielsen data covers the majority of in-store retailing sales in the country, the data is not exclusive and we do not have the complete air conditioner transaction records (e.g. the online purchase records). This is unlikely to threaten the key conclusions of our research unless the estimated effect is corrected with the spatial or temporal pattern of the data availability, which is implausible. However, such partial

exclusion makes it difficult to provide estimates on the increased purchases at a more aggregated (e.g., county) level, which matters in quantifying the total electricity saved. Moreover, the retailer scanner data only record the sales at the store level but does not enable the identification of consumer types. As commercial entities can make their purchase decision differently compared to private households, it is critical to discern the effect of short-run weather on each for more targeted policy implications. We leave these for future research that draws on more temporally refined data and/or records covering more complete channels of sales.

This relationship deserves both attention from academic researchers and policymakers. Previous research on the energy efficiency gap mostly leverages nudges and other types of economic incentives to increase the adoption of energy-efficient products[3,31]. Our results open many possibilities to incorporate temperature fluctuations in designing nudge or other incentive schemes as consumers are more likely to pay attention to Energy Star products under certain circumstances, which could significantly increase the effectiveness of current nudge or incentive programs. Similarly, businesses can incorporate temperature fluctuation to develop weather-based marketing strategies to better promote energy-efficient products[32], particularly targeting the population identified as potentially more responsive in our tests of heterogeneous effects. Furthermore, policymakers shall integrate the consideration of consumer behavior when estimating the loss of climate change while exploring the opportunities of behavior change as a low-hanging fruit of GHG mitigation. Given that the heat exposure will be continuously increasing not only in the United States but also at the global level[33], such demand-end strategies can play a critical role in leapfrogging over the intensive energy consumption towards higher adoption of energy efficiency products, particularly where the demand of air conditioning will be rapidly increasing such as the BRIC countries[34].

## Methods

### Data

We use transaction-level air conditioner purchase data from the Nielsen Retail Scanner Dataset. This dataset contains weekly pricing and volume of products in various categories sold from ~30,000–50,000 (depending on the year) participating grocery, drug, mass merchandiser, and other stores in all US markets. We extracted around 1,900,000 purchase records of air conditioners in more than 13,600 stores from approximately 2100 counties in 48 states from 2006 to 2019. We then identified the models with Energy Star labeling by conducting an online search of the product information of more than 200 models that emerged in the dataset.

The geographical identifier of the stores is available in the Retail Scanner Dataset at the county level, enabling us to link it with the ground-station level meteorological records from Global Surface Summary of the Day (GSOD)[20]. Each store is matched with all the climate stations within a 100 km buffer of the county where it is located in. As GSOD provides meteorological data at the daily level, we calculate the weekly average of all the matched stations. The meteorological data in the previous week and used for matching and regression analysis since contemporaneous meteorological records may contain information on the weather ahead of the early days of the present week, and it also takes time for consumers to take into consideration purchasing a durable good such as air conditioner in response to the ambient temperature.

### Empirical strategy

We conduct transaction-level regression analysis to investigate if the weather factors affect the purchase of air conditioners labeled with Energy Star. Following a previous study that conducts the regression of weather indicators on the dummy variable of transaction outcomes[9],

we construct the specification as

$$I(\text{ES}_{irt}) = \sum_j \beta_{1j} f(\text{temp}_{rt}) + \boldsymbol{\delta} \mathbf{W_{rt}} + \beta_2 \text{price}_{irt} + \mu_{rw} + \gamma_{smT} + \varepsilon_{irt} \quad (1)$$

where $I(\text{ES}_{irt})$ is a dummy denoting whether the $i$th transaction in $r$th county on week $t$ is Energy Star-certified. $\sum \beta_{1j} f(\text{temp}_{rt})$ are a set of bins indicating whether the weekly average temperature in $r$th county on week $t$ falls into a specific interval. After testing different settings of default groups, we found that the probability of purchasing Energy Star air conditioners increases as temperature deviates from 20 to 22 °C, which is the default group in the final regression. $\text{price}_{irt}$ is the price of the transaction. The vector $\boldsymbol{W}_{rt}$ contains other meteorological variables as controls including precipitation, wind speed, and relative humidity and its square. Literature has indicated that heat stress affects human behavior not only via temperature but also via humidity in that a relative humidity that is too high or too low would add to thermal uncomfort[21,35]. We also include a series of fixed effects, including the county*week-of-year fixed effect $\mu_{rw}$, and the state*year*-month fixed effect $\gamma_{smT}$. The combination of these fixed effects captures the confounding effects of heterogeneous seasonality and socio-economic dynamics (such as electricity price) at the county/state level. Specifically, the geographic distribution of Energy Star products may be different from the non-Energy Star products as a result of the weather in the long run. If this holds, the effect of weather can be attributed to the accessibility of the Energy Star products in particular stores instead of the psychological mechanisms of the consumers. The two sets of fixed effects included in the model capture such differences across counties and over time in the long run, and thus enable us to isolate the effect of the short-run weather on consumer decisions. The error term is $\varepsilon_{irt}$. As each transaction may contain more than 1 air conditioner, we weight the regression using the units of sales in each transaction.

The key variable of interest is temperature as we assume that deviation from thermal comfort would increase the significance of temperature in consumer choices and make the Energy Star models preferable due to potential energy cost saving in the future. We expect $\beta_{1j}$ to be significant if the temperature affects the valuation of energy efficiency. Concerning the choice of model, a Probit or Logit model would be ideal for a dummy dependent variable such as in our case. However, rarely do such models converge after including the long list of fixed effects which are also essential in controlling for unobservable against possible confounding effects. Therefore, we estimate the model based on the generalized least square (GLS) method as a compromise following the previous studies that investigate the effect of weather on purchase decisions with a similar form of data[9,10] using the reghdfe command in Stata. In order to facilitate the tests of heterogeneous effects, we also conduct the same regression replacing $\sum \beta_{1j} f(\text{temp}_{rt})$ with the aggregated cooling degree days (CDDs) and Heating degree days (HDDs) of the week. When testing the heterogeneous effects, we interact the background CDDs/HDDs, state-level electricity price, and other variables of interest with both the summed CDDs and HDDs of the week as well as other meteorological indicators. The descriptive statistics of the variables adopted in the analysis can be found in Supplementary Table 1. The code and available datasets are included in the Github repository[36].

### Reporting summary

Further information on research design is available in the Nature Research Reporting Summary linked to this article.

## Data availability

The GSOD data are accessed using GSODR package in R and can also be retrieved from https://www1.ncdc.noaa.gov/pub/data/gsod/. The Retail Scanner data is provided by Nielsen Company and restricted

by non-disclosure terms of use but can be purchased from Nielsen. According to the official website at https://www.chicagobooth.edu/research/kilts/datasets/nielseniq-nielsen/subscribing, accessing the data would first require a subscription contract between a researcher's institution and Chicago Booth. (Researcher(s)' own analyses calculated (or derived) based in part on data from Nielsen Consumer LLC and marketing databases provided through the NielsenIQ Datasets at the Kilts Center for Marketing Data Center at The University of Chicago Booth School of Business. The conclusions drawn from the NielsenIQ data are those of the researcher(s) and do not reflect the views of NielsenIQ. NielsenIQ is not responsible for, had no role in, and was not involved in analyzing and preparing the results reported herein.) The metrics of climate attitude are available in the supplementary material of the study "Geographic variation in opinions on climate change at state and local scales in the USA"[26]. The annual state-level electricity price is obtained from the U.S. Energy Information Administration website at https://www.eia.gov/electricity/data/state/. The county-level socioeconomic and demographic characteristics are available in the American Community Survey from https://www.census.gov/geographies/mapping-files/time-series/geo/tiger-data.html. The support of the democratic party in county presidential election returns of 2000–2020 comes from at https://dataverse.harvard.edu/dataset.xhtml?persistentId=doi:10.7910/DVN/VOQCHQ. Source data are provided with this paper and also available from the github repository https://github.com/hepannju/Does-the-weather-change-the-energy-efficiency-gap (https://doi.org/10.5281/zenodo.7092652). Source data are provided with this paper.

## Code availability

The major data processing and all the regression analysis are conducted in Stata 16.0. The download of GSOD data and figure production is conducted in R studio (based on R 4.0.2). All custom code is available from https://github.com/hepannju/Does-the-weather-change-the-energy-efficiency-gap (https://doi.org/10.5281/zenodo.7092652).

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

## Acknowledgements

P.H. acknowledges support from the National Natural Science Foundation of China under Young Scholar Program Grant #71904098, this is a Cardiff EARTH CRediT Contribution 3. P.L. acknowledges support from the U.S. Department of Agriculture (USDA) National Institute of Food and Agriculture, Hatch Regional project accession number 1014661. Y.L.Q. acknowledges support from National Science Foundation Award #1757329.

## Author contributions

P.H., Y.Q., and P.L. designed the study. P.L. and L.L. processed the data. P.H. performed the analysis and drew the figures. P.H., Y.Q., and P.L. participated in discussing the results and contributed to writing the manuscript.

## Competing interests

The authors declare no competing interests.
