## [Peer review file · Nature Communications]

REVIEWER COMMENTS

Reviewer #1 (Remarks to the Author):

This manuscript aims at estimating the effect of weekly weather fluctuations on the purchase of energy-efficient air conditioners. The authors' research question is relevant and motivated. It is pretty interesting the application of Busse et al. (2015)'s analysis to the trade-off between energy-efficiency and adaptation to climate-change. The analysis is well conducted, and the paper is quite well written in most sections. However, there are some concerns that have to be addressed.

Major issues

1. In the light of their results, I believe the paper would benefit from a more extended discussion of how weekly weather affect the decision of purchasing an Energy Star air conditioners. In Busse et al. (2015) this is more immediate, and they explained it extensively. Indeed, both convertibles and four-wheel-drive vehicles have attributes (e.g. convertible roof) that are strictly correlated to weather. This then correlates with the main psychological and behavioural biases they identify.

Here, the good is air conditioner, which, independently from their energy efficiency attribute, tends to be purchased when agents are subject to a warmer climate (or after positive temperature shocks). Assuming Energy Star air conditioners are more expensive (are they?), the questions are then: why do an external idionsyncratic shocks like weather make a buyer purchase more likely an energy efficient air-conditioning (even when it is more expensive)? And then, what is the correlation between air conditioner's energy efficiency and weather? I guess the answer to the second question is the operational cost of the air conditioner. Can we then say that projection bias and salience correlates with the fact that American buyers might value more the operational cost rather than the capital cost of an air conditioner? If there are studies that document this is the case in US (even for other energy appliances), I think they might help the discussion.

2. Remaining in the spirit of Busse et al. (2015), a potential further mechanism may be worth to explore is how the supply, i.e. prices, react to temperature shocks. Whereas cars' prices are independent from daily temperature, air conditioners' price might be more likely to fluctuate. For instance, during warm weeks Energy Star air conditioner might be on sale. If there is no effect, this is even more confirmatory of the main role of psychological biases in agents' behaviour.

3. I believe a reader would benefit from an interpretation of the coefficients the authors find. Even though they describe the trend of Energy Star purchase along temperature bins, as well as the significance of their coefficients, no magnitude interpretation is provided.

4. While I believe it is clearly difficult to isolate which mechanism is taking place, I think the heterogeneity results can partially help in this sense. Do the results for previous years' CDDs provide intuition that "salience" is occurring?

Minor issues

5. In the introduction the authors write "pay limited attention to the attribute of energy efficiency in their purchasing decisions", and they cite Salee et al. (2014). However, Rapson (2014) shows that consumers are forward-looking and value the stream of future savings derived from energy efficiency. Rapson (2014) particularly refers to air conditioners' purchase. Hence, authors' statement is not totally true. I would say that the evidence is conflicting.

6. In all Figures along the manuscript the y-axis is labelled as "Probability, Energy Star Model". I believe it would be better to clarify whether the marginal effect they are shown is in percentage points (%) or not.

7. In the regression specification described in "Methods", it is indicated a λ_w - which I interpret as a dummy for week. However, there is no description in the text about this covariate, as well as it is not present in the Supplementary Tables. Is it used in the regressions?

8. I appreciate the use of state x year x month fixed effect. Did the authors also test for less stringent time fixed effects? This is to check that the results remain consistent if they add more variability

Reference:

Rapson, D. (2014). Durable goods and long-run electricity demand: Evidence from air conditioner purchase behavior. *Journal of Environmental Economics and Management*, 68(1), 141-160.

Reviewer #2 (Remarks to the Author):

The paper tackles an interesting question, namely whether temperature deviations affect the uptake of energy-efficient AC units.

While the authors provide an interesting background, it would be interesting to know if similar study have previously addressed this question or whether it is a first-in-its-kind analysis.

Another question is whether the survey only covers purchases from private households, or it could also include transaction from businesses and commercial activities.

Moving to the more substantial points, something which is not fully clear to me is whether such purchases refer to households who (A) already own an AC (i.e. is it a replacement/retrofitting), whether it is instead (B) purchases leading to a growing penetration of the appliance or (C) whether this information is not inferrable from the survey data in question. I appreciate that in the US AC penetration rate is 90%, so most instances are likely to be referring to case (A), but this should at least be made clear from the beginning.

Namely, in cases B and C, it is misleading (or, at least, not possible) to conclude that “1°C increase and decrease from 21°C would lead to a reduction of total energy expenditure by 35.46 and 17.73 million dollars (0.13% and 0.06% of the annual total energy expenditure on air conditioning)”.

This is because it is impossible to separate the components of (i) decreasing marginal energy expenditure (USD/kWh of cooling energy consumed) due to the use of higher efficiency appliances and (ii) increasing absolute consumption due to AC uptake of households which previously did not own AC. Indeed, my suspect is that on national terms (ii) might well outweigh (i).

Relatedly, you should remark that the conclusion of energy expenditure saving refers to a counterfactual situation of constant climate. If we consider growing intensive margin of AC due to a warming climate, a third effect (iii) might then outweigh both (i) and (ii).

Figure 3: it is not fully clear what are / how are defined groups 1 2 and 3 in the x-axis.

###

Overall, before being able to make a decision, I would kindly ask the authors to address and comment upon the issues mentioned above.

Reviewer #3 (Remarks to the Author):

This paper explores the effect of short-run weather fluctuations in narrowing the energy efficiency gap by focusing on air conditioner purchases in the US from 2006 to 2009. The main results show that deviating from the comfort temperature range significantly increases Energy Star product purchases relative to non-efficient ones and the results remain robust after several robustness checks. Also, the stimulation effect of temperature would be amplified by higher state-level electricity price, fewer historical Cooling Degree Days (CDDs), higher household income, higher educational levels, more intensive concern about climate change, and higher support of Democratic Party. This paper has a good idea about the setting of independent variables and has completeness of empirical tests. Here are some comments I would incorporate into the paper before publication.

1. It seems that the coefficient plot does not match the regression results table. For example, both Figure 1-a and Table S1-(1) show the baseline results for the full sample. In Table S1-(1), the coefficient of temperature interval [$<0^{\circ}\text{C}$] is -0.097, but in Figure 1-a, the coefficient of temperature interval [$<0^{\circ}\text{C}$] is lower than -0.1 (about -0.16). In fact, the coefficient plot of "The effect of temperature on Energy Star air conditioner purchase" (Figure 1) matches the regression results table of "The effect of temperature on Energy Star Telephone purchase" (Table S1-(5)(6)) well. Figure S2 does not match either regression results table. In this paper, the software used for figure production (R studio) is different from the software used for regression analysis (Stata 16.0). I guess there is something wrong when migrating the regression results, and this needs the author to check.
2. Model. This paper conducts linear probability models. However, for dummy dependent variables, is it more appropriate to use Probit or Logit model? Can you explain the reasons for choosing linear probability models and use the results of the Logit and Probit models as robustness checks?
3. Data. In line 105-107, since the common prices for air conditioners have been listed as \$150-\$750 and deleted records with transaction prices below \$100, why not delete records above a certain dollar?
4. In Table-S1(3)(4), the author uses the temperature deviation from 21°C as regressors to estimate the marginal effects. How about the nonlinear effects of temperature deviation? Authors could try to use linear and quadratic terms of temperature deviation as regressors. Similarly, incorporating the linear and quadratic terms of meteorological variables into the model can also be used as robustness checks.

5. Robustness checks. What is the basis for selecting a comfort zone of 20-22 degrees C and a calculated threshold of 70 degrees Fahrenheit for CDDs and HDDs for this paper? It is recommended to consider robustness checks for different comfort intervals and thresholds and thus investigate whether the results depend on the setting of both. Also, are the results robust to different clustering choices?
6. Socioeconomic characteristics. Do socioeconomic characteristics include age and household characteristics, etc.? The impact of demographics on ENERGY STAR air conditioner purchases may be substantial and bias the estimates if demographics are not randomly assigned among locations.
7. In line 106-108, it is not clear on the explanation of how the control of the week's weather indicators addresses the problems that may be caused by the geographic distribution of Energy Star differing from that of non-Energy Star air conditioners. For example, why not control for the geographic location of where air conditioners are purchased, etc.
8. The summary statistics table is needed.
9. Please check the sequence numbers of Figure and Table in the text. For example, in page 6, all "Figure 2" should be revised to "Figure 3".
10. In Equation (1), W_{rt} should be revised to $W_{rt} \delta$.
11. In line 126-128, why choose one month before the transaction and three weeks after the transaction, instead of the same period?
12. Figure 3. "Bckground climate", is this a typo here?

REVIEWER COMMENTS

Reviewer #1 (Remarks to the Author):

This manuscript aims at estimating the effect of weekly weather fluctuations on the purchase of energy-efficient air conditioners. The authors' research question is relevant and motivated. It is pretty interesting the application of Busse et al. (2015)'s analysis to the trade-off between energy-efficiency and adaptation to climate-change. The analysis is well conducted, and the paper is quite well written in most sections. However, there are some concerns that have to be addressed.

We thank the reviewer for careful review of the manuscript and many valuable comments. We took time and addressed each comment carefully in the revision. We think the current manuscript is significantly improved and this version has addressed all the important concerns. Below we provide response to each comment in detail (the reviewer's comments are in blue and our responses starting with **Response** are in black).

Major issues

1. In the light of their results, I believe the paper would benefit from a more extended discussion of how weekly weather affect the decision of purchasing an Energy Star air conditioners. In Busse et al. (2015) this is more immediate, and they explained it extensively. Indeed, both convertibles and four-wheel-drive vehicles have attributes (e.g. convertible roof) that are strictly correlated to weather. This then correlates with the main psychological and behavioural biases they identify.

Here, the good is air conditioner, which, independently from their energy efficiency attribute, tends to be purchased when agents are subject to a warmer climate (or after positive temperature shocks). Assuming Energy Star air conditioners are more expensive (are they?), the questions are then: why do an external idiosyncratic shocks like weather make a buyer purchase more likely an energy efficient air-conditioning (even when it is more expensive)? And then, what is the correlation between air conditioner's energy efficiency and weather? I guess the answer to the second question is the operational cost of the air conditioner. Can we then say that projection bias and salience correlates with the fact that American buyers might value more the operational cost rather than the capital cost of an air conditioner? If there are studies that document this is the case in US (even for other energy appliances), I think they might help the discussion.

Response: We thank the reviewer for the suggestion. As shown in the descriptive statistics added in this revision (in Table S1 included below), the Energy Star air conditioners are indeed more expensive. We also agree with the reviewer that the interpretation of the correlation between the weather shocks and the purchase of Energy Star air conditioners that we found is likely that the consumers tend to value more the operational cost due to the reduction of thermal comfort. The evidence is mixed on whether the consumers are forward-looking or myopic when purchasing durable goods like air conditioners as suggested in your point 5, and thus there is no explicit

conclusion whether the American buyers value more the operational cost rather than the capital cost. It is worth pointing out this possibility for future studies, and we have included this point in the third paragraph of the robustness check when we mention the two potential mechanisms:

“... CDDs and HDDs remain stable. Provided that the Energy Star air conditioners are of higher prices on average (Table S1), the two mechanisms are likely to enlarge the value of consumers on the operational cost rather than the capital cost of durable goods. To date, evidence is mixed on whether the consumers are forward-looking or myopic in durable good purchases^{7,8}, yet our findings suggest that short-run weather changes may add to the rationality of consumers.”

Table S1 Descriptive statistics

	N	Mean	SD	Min	Max
Transaction level	215419				
Price (\$)	8	175.44	87.114	0.01	699.99
Energy Star	160834				
Non- Energy Star	4	170.196	94.762	0.01	699.99
County level	545854	190.892	56.31	0.01	649.99
Temperature (°C)	194509				
	2	19.841	5.011	-21.036	40.5
CDDs	194509				
	2	19.816	26.007	0	244.3
HDDs	194509				
	2	35.818	45.042	0	531.05
Wind speed (m/s)	192058				
	5	2.914	0.934	0	21.8
Precipitation (mm)	194465				
	5	2.351	3.067	0	67.157
Relative Humidity (%)	188361				
	5	66.267	11.458	0.3	100
Residential electricity price (\$/kwh)	215419				
	7	12.456	2.786	6.88	15.803
CDDs, 1996-2005	215162				
	0	8.753	8.337	0	70.325
HDDs, 1996-2005	215162				
	0	125.552	34.171	9.454	216.324
Median annual household income (\$)	167422	6.70E+0	1.90E+0	2.00E+0	1.40E+0
Population with bachelors above 25 (%)	1	4	4	4	5
	167422				
	1	0.338	0.109	0.054	0.753
Population of white (%)	167422				
	1	0.733	0.179	0.155	0.998
Median age (Years)	167422				
	1	39.041	3.776	22.7	59.1

	167422				
Median number of rooms	1	5.557	0.696	3.3	7.7
	167422				
Owner : Renter (Ratio)	1	2.04	0.998	0.236	8.599
	167422				
Electricity as the heating fuel (%)	1	0.234	0.197	0.012	0.975
Population believing climate change happening	215176	5	66.258	5.807	43.9
Population believing climate change will harm the US (%)	215176	5	55.176	4.979	41.6
Population worried about climate change (%)	215176	5	55.597	6.694	37.9
Population supporting renewable energy standards (%)	215176	5	57.659	4.387	40.1
Population supporting regulation of CO ₂ as a pollutant (%)	215176	5	69.246	3.836	49.8
	215114				
Support Democrats in election (%)	3	0.551	0.158	0.048	0.925

2. Remaining in the spirit of Busse et al. (2015), a potential further mechanism may be worth to explore is how the supply, i.e. prices, react to temperature shocks. Whereas cars' prices are independent from daily temperature, air conditioners' price might be more likely to fluctuate. For instance, during warm weeks Energy Star air conditioner might be on sale. If there is no effect, this is even more confirmatory of the main role of psychological biases in agents' behaviour.

Response: We thank the reviewer for this suggestion. Following the reviewer's suggestion, we further investigated how different meteorological factors affect air conditioner prices and included the results in Table S8 in the Supplementary Information (also presented below). Our results show that CDDs and HDDs do have a significant impact on the price of Energy Star & non-Energy Star air conditioners. For the full sample, the Energy Star products tend to be cheaper when the weather is either hotter or colder, indicating that there may be a larger promotion of these products. Nevertheless, when we exclude the price outliers, Energy Star products are of higher prices with increased CDDs. Since we have controlled for the price with a weekly variation in the main regression, the effect of price fluctuation on the purchase decision is controlled and would not threaten our key findings. To address the reviewer's comment, we have added a related discussion in the section of Robustness check:

"It is possible that the weather affects the sales of air conditioners via the change of price, e.g. promoting Energy Star air conditioners with larger discounts compared with non-Energy Star products as CDDs/HDDs increase. We test this possibility by interacting the dummy of Energy Star in the transaction with the meteorological variables, and run a regression of these variables on the transaction price (in log form). As shown in Table S8, the Energy Star products are of a lower price when the weather is either hotter or colder in the full sample, but the significance turns to the opposite sign for CDDs after the exclusion of the price outliers. Although it is hard to reach a meaningful conclusion based on the regression results, we have controlled for the

price with a weekly variation in the main regression so that the effect of price fluctuation on the purchase decision is controlled and would not threaten our key findings. ”

Table S8 The effect of weather on air conditioner price

	(1) Main	(2) Price Outlier
EnergyStar	0.285*** (0.048)	0.203*** (0.053)
CDD	-0.001 (0.001)	-0.004*** (0.001)
EnergyStar*CDD	-0.010*** (0.001)	0.006*** (0.002)
HDD	0.002*** (0.001)	0.001*** (0.000)
EnergyStar*HDD	-0.008*** (0.001)	-0.003*** (0.001)
Wind speed	0.004 (0.002)	0.012*** (0.003)
EnergyStar*Wind speed	-0.016*** (0.005)	-0.039*** (0.007)
Precipitation	-0.000 (0.000)	-0.000 (0.001)
EnergyStar*Precipitation	0.003*** (0.001)	0.001 (0.001)
Relative humidity	-0.036*** (0.009)	-0.024*** (0.008)
EnergyStar*Relative humidity	-0.016 (0.014)	-0.012 (0.015)
Relative humidity ²	0.003*** (0.001)	0.003*** (0.001)
EnergyStar*Relative humidity ²	0.001 (0.001)	-0.000 (0.001)
Fixed effects		
County*week	Y	Y
State*year*month	Y	Y
N	1871472	1564506
R ²	0.280	0.276

Notes: Standard errors in parentheses are clustered to store level. Results weighted by the units sold per transaction.
* p<0.1, ** p<0.05, *** p<0.01. The CDD, HDD, and relative humidity multiplied by 10.

3. I believe a reader would benefit from an interpretation of the coefficients the authors find. Even though they describe the trend of Energy Star purchase along temperature bins, as well as the significance of their coefficients, no magnitude interpretation is provided.

Response: We thank the reviewer for this comment. We included interpretations for the regressors of temperature deviation from 21°C in the second paragraph of the section “Effects of weather on Energy Star Air conditioner Purchase” in the previous manuscript. We agree that more interpretations regarding the magnitude of the coefficients should be included to provide the readers with a clear picture regarding the overall effects. Following the reviewer’s suggestion, we have added the following interpretations in the first paragraph of the same section below:

“... the United States¹. The probability of purchasing an Energy Star air conditioner increases by more than 2% when the temperature shift raises to 24-32°C, and by more than 5% when the temperature exceeds 32°C. By contrast, colder weather encourages such transactions by 0.7%-3%. Although the air ...”

4. While I believe it is clearly difficult to isolate which mechanism is taking place, I think the heterogeneity results can partially help in this sense. Do the results for previous years' CDDs provide intuition that "salience" is occurring?

Response: We thank the reviewer for this comment. The results for previous years’ CDDs suggest that it is possible that the residents in milder climates are more sensitive to the change of weather and the importance of energy saving becomes more salient as the reviewer suggested. Alternatively, it can be due to that the consumers in hotter places resort to central cooling which may be more effective and efficient with intensive cooling demand. Following the reviewer’s suggestion to link the heterogeneity results and the mechanisms, we have revised the explanation of the result of background CDDs to incorporate the reviewer’s suggestion in the first paragraph of the section “Heterogeneous effects of temperature” as below:

“... Results show that individuals are more affected in states with higher average electricity prices (Figure 3-a, and Column 1, Table S10). They also tend to have a stronger response with fewer background CDDs in the previous years (Figure 3-b, and Column 2, Table S10). The HDDs, however, show a non-monotonic effect on the extent of response. The response to temperature first rises and then declines as the background HDDs increase (Figure 3-c, and Column 3, Table S10). This may imply the existence of salience as individuals are more sensitive to temperature change when the background temperature is milder so that the importance of energy saving starts to emerge. Another possible reason is that in more drastic climate, consumers tend to resort to other cooling/heating appliances rather than window air conditioners such as central cooling/heating systems, heat pump, etc.”

Minor issues

5. In the introduction the authors write "pay limited attention to the attribute of energy efficiency in their purchasing decisions", and they cite Salee et al. (2014). However, Rapson (2014) shows that consumers are forward-looking and value the stream of future savings derived from energy efficiency. Rapson (2014) particularly refers to air conditioners' purchase. Hence, authors' statement is not totally true. I would say that the evidence is conflicting.

Response: We thank the reviewer for the comment and suggestion on the reference. To address the reviewer’s comment, we have revised the statement with Rapson (2014) cited:

“... In addition, such predictions need to be accurate and consistent regardless of time and place, which has been challenged by recent evidence from behavioral economics where consumers could heavily discount fuel economy in the future ^{2,3}, and conflicting evidence on whether consumers pay limited attention to the attribute of energy efficiency in their purchasing decisions ^{4,5}. As a result, ...”

6. In all Figures along the manuscript the y-axis is labelled as "Probability, Energy Star Model". I believe it would be better to clarify whether the marginal effect they are shown is in percentage points (%) or not.

Response: We thank the reviewer for this suggestion and we apologize for the ambiguity. Following the reviewer’s suggestion, we have revised the y-axis label in Figure 1 to percentage points. We have also updated Figure 2, Figure 3, Figure S2, and Figure S3 to label axes clearly.

7. In the regression specification described in "Methods", it is indicated a lambda_w - which I interpret as a dummy for week. However, there is no description in the text about this covariate, as well as it is not present in the Supplementary Tables. Is it used in the regressions?

Response: We thank the reviewer for pointing this out. The variable lambda_w is not used in the regression (was used in an earlier version). We have dropped lambda_w from Equation 1.

8. I appreciate the use of state x year x month fixed effect. Did the authors also test for less stringent time fixed effects? This is to check that the results remain consistent if they add more variability

Response: We thank the reviewer for this suggestion. Following the reviewer’s suggestion, we have included more tests with alternative combinations of less stringent fixed effects in Table S3 and below in the revision. As shown by the coefficients of the terms of CDD and HDD, the effect of temperature remains robust in both statistical significance and magnitude. We also included an interpretation of this table in the section “Robustness checks”:

“In case our results are sensitive to the setting of fixed effects, we switch to alternative combinations of fixed effects and rerun the major analysis on CDDs and HDDs. As shown in Table S7, the coefficients of both terms remain significant with similar magnitude, indicating the robustness of our results.”

Table S7 The effect of temperature on Energy Star air conditioner purchase with alternative combinations of fixed effects

	(1)	(2)	(3)	(4)	(5)
	EnergyStar	EnergyStar	EnergyStar	EnergyStar	EnergyStar
CDD	0.002*** (0.001)	0.003*** (0.001)	0.003*** (0.001)	0.003*** (0.001)	0.002*** (0.001)
HDD	0.001*** (0.000)	0.002*** (0.000)	0.002*** (0.000)	0.001*** (0.000)	0.002*** (0.000)

Wind speed	0.006*** (0.002)	0.007*** (0.002)	0.006*** (0.002)	0.006*** (0.002)	0.004** (0.002)
Precipitation	-0.002*** (0.000)	-0.002*** (0.000)	-0.002*** (0.000)	-0.002*** (0.000)	-0.002*** (0.000)
Relative humidity	-0.014*** (0.005)	-0.015*** (0.005)	-0.015*** (0.005)	-0.013*** (0.005)	-0.010** (0.004)
Relative humidity ²	0.002*** (0.000)	0.002*** (0.000)	0.002*** (0.000)	0.002*** (0.000)	0.001*** (0.000)
Inprice	0.099*** (0.005)	0.100*** (0.005)	0.100*** (0.005)	0.100*** (0.005)	0.096*** (0.005)
Fixed effects					
County*week		Y	Y	Y	Y
County*month	Y				
State*year					Y
State*month				Y	
State	Y	Y	Y		
Year	Y	Y	Y	Y	
Month			Y		Y
Week	Y				
N	1879080	1871984	1871984	1871982	1871981
R ²	0.128	0.144	0.144	0.146	0.169

Notes: Standard errors in parentheses are clustered to store level. Results weighted by the units sold per transaction.

* p<0.1, ** p<0.05, *** p<0.01. The CDD, HDD, and relative humidity multiplied by 10.

Reference:

Rapson, D. (2014). Durable goods and long-run electricity demand: Evidence from air conditioner purchase behavior. *Journal of Environmental Economics and Management*, 68(1), 141-160

Reviewer #2 (Remarks to the Author):

The paper tackles an interesting question, namely whether temperature deviations affect the uptake of energy-efficient AC units.

We thank the reviewer for careful review of the manuscript and many valuable comments. We took time and addressed each comment carefully in the revision. Below we provide response to each comment in detail (your comments are in blue and our responses starting with Response are in black).

While the authors provide an interesting background, it would be interesting to know if similar study have previously addressed this question or whether it is a first-in-its-kind analysis.

Response: We thank the reviewer for this comment. There is a branch of literature examining how the physical environment impacts the purchase decisions by affecting consumers' decision-making process summarized in the second paragraph of the introduction section. Nevertheless, we firstly explore whether the environmental factors change the recognition of the feature of energy saving in appliance purchases to our knowledge. We really appreciate the reviewer's point which helps us to improve the positioning of the contribution of this study. In the revision, we have incorporated the reviewer's point in the last paragraph of the same section and stated our contribution as below:

"... for environmental benefit. To our knowledge, this is the first study exploring whether the environmental factors change the recognition of energy efficiency in appliance purchases. Our results indicate a ..."

Another question is whether the survey only covers purchases from private households, or it could also include transaction from businesses and commercial activities.

Response: We thank the reviewer for this point. As our major analysis focuses on the retailer scanner data that captures the sales of air conditioners from the stores, it is unlikely that transactions from businesses and commercial activities are included. The decision-making process of commercial entities could be different from that of private households. Additional data is needed to address the difference between private households' and business entities' responses to the short-run weather fluctuations. To address the reviewer's comment, we have added this limitation in the third paragraph of the section "Discussion" as below:

"... the total electricity saved. Moreover, the retailer scanner data only record the sales at the store level but do not enable the identification of consumer types. As the commercial entities can make their purchase decision differently compared to private households, it is critical to discern the effect of short-run weather on each for more targeted policy implications. We leave these for future research ..."

Moving to the more substantial points, something which is not fully clear to me is whether such purchases refer to households who (A) already own an AC (i.e. is it a replacement/retrofitting),

whether it is instead (B) purchases leading to a growing penetration of the appliance or (C) whether this information is not inferrable from the survey data in question. I appreciate that in the US AC penetration rate is 90%, so most instances are likely to be referring to case (A), but this should at least be made clear from the beginning.

Namely, in cases B and C, it is misleading (or, at least, not possible) to conclude that “1°C increase and decrease from 21°C would lead to a reduction of total energy expenditure by 35.46 and 17.73 million dollars (0.13% and 0.06% of the annual total energy expenditure on air conditioning)”.

This is because it is impossible to separate the components of (i) decreasing marginal energy expenditure (USD/kWh of cooling energy consumed) due to the use of higher efficiency appliances and (ii) increasing absolute consumption due to AC uptake of households which previously did not own AC. Indeed, my suspect is that on national terms (ii) might well outweigh (i).

Relatedly, you should remark that the conclusion of energy expenditure saving refers to a counterfactual situation of constant climate. If we consider growing intensive margin of AC due to a warming climate, a third effect (iii) might then outweigh both (i) and (ii).

Response: We thank the reviewer for this comment. In our data, we cannot distinguish whether it's a new purchase or it's a replacement. So the option it's (C). However, since the penetration rate is ranging from 82% to 88% during 2005-2020⁶, it's more likely the air conditioner is a replacement.

Our current calculation is all based on replacing the existing transactions with Energy Star products as the temperature changes. In this sense, we don't address how the climate would make a difference on the total amount of air conditioner ownership by promoting new purchases. Future research can explore this topic in detail when the data on the nature of individual purchase (replace or new purchase) is available to calculate the net effect of the temperature impacts on the overall energy consumption through air conditioner usage. We understand the reviewer's concerns that if hotter/colder days also encourage new purchases of Energy Star Air Conditioners that would have not taken place in a milder climate, the benefit of energy saving will be at least partially balanced out. Therefore, we have made it clear that the current calculation only addresses the current transactions and our comparison is based on a benchmark when we do not address the impact of the temperature on the probability of purchasing an energy star certified air conditioner among the consumers. We have also indicated this limitation of the estimation in the first paragraph of the Discussion section:

“...Provided that the annual average air conditioner shipped in the U.S. during 2006-2019 is approximate 4.44 million²⁸, our estimations imply that Energy Star conditioner purchases will increase by 17760 and 8880 with 1°C increase and decrease from 21°C, respectively, as a replacement of current transactions. ... In this way, 1°C increase and decrease from 21°C would help to save electricity use by 292.4 and 146.2 billion Btu (60,730 and 30,365 tons of CO_{2e}, provided that the air conditioning accounts for 731 trillion Btu of the household energy consumption in 2015³⁰) compared with the status quo of energy consumption from air

conditioners in a counterfactual situation of constant climate, respectively, as a back-of-envelope estimation. ... It would also lead to a reduction of total energy expenditure by 35.46 and 17.73 million dollars (0.13% and 0.06% of the annual total energy expenditure on air conditioning) at the national level compared to the scenario when we assume the temperature would not impact the probability of purchasing an energy star certified air conditioner, respectively. This benefit may be partially canceled out if the temperature change also encourages new purchases of Energy Star products among those who would have lived without air conditioning in a constant climate. Quantitative estimation on the magnitude of this effect is not available given the limitation of our data. Nevertheless, it could be small provided that the penetration rate of air conditioning has hit 88% in 2020³⁰.”

Figure 3: it is not fully clear what are / how are defined groups 1 2 and 3 in the x-axis.

Response: We thank the reviewer for pointing this out and apologize for the ambiguity. We have replaced the x-axis labels in Figure 3 to be “low”, “medium”, and “high”, and added a description in the caption to explain how the grouping is conducted as below:

“...the 95% confidence intervals. The groups are divided using tertiles of the variable that indicates potential heterogeneous preference.”

###

Overall, before being able to make a decision, I would kindly ask the authors to address and comment upon the issues mentioned above.

Response: We the reviewer again for many valuable comments which greatly help us to improve the quality of this study. We think the current manuscript is significantly improved and this version has addressed all the important concerns.

Reviewer #3 (Remarks to the Author):

This paper explores the effect of short-run weather fluctuations in narrowing the energy efficiency gap by focusing on air conditioner purchases in the US from 2006 to 2009. The main results show that deviating from the comfort temperature range significantly increases Energy Star product purchases relative to non-efficient ones and the results remain robust after several robustness checks. Also, the stimulation effect of temperature would be amplified by higher state-level electricity price, fewer historical Cooling Degree Days (CDDs), higher household income, higher educational levels, more intensive concern about climate change, and higher support of Democratic Party. This paper has a good idea about the setting of independent variables and has completeness of empirical tests. Here are some comments I would incorporate into the paper before publication.

We thank the reviewer for careful review of the manuscript and many valuable comments. We took time and addressed each comment carefully in the revision. We think the current manuscript is significantly improved and this version has addressed all the important concerns. Below we provide response to each comment in detail (your comments are in blue and our responses starting with **Response** are in black).

1. It seems that the coefficient plot does not match the regression results table. For example, both Figure 1-a and Table S1-(1) show the baseline results for the full sample. In Table S1-(1), the coefficient of temperature interval [$<0^{\circ}\text{C}$] is -0.097, but in Figure 1-a, the coefficient of temperature interval [$<0^{\circ}\text{C}$] is lower than -0.1 (about -0.16). In fact, the coefficient plot of “The effect of temperature on Energy Star air conditioner purchase” (Figure 1) matches the regression results table of “The effect of temperature on Energy Star Telephone purchase” (Table S1-(5)(6)) well. Figure S2 does not match either regression results table. In this paper, the software used for figure production (R studio) is different from the software used for regression analysis (Stata 16.0). I guess there is something wrong when migrating the regression results, and this needs the author to check.

Response: We thank the reviewer for pointing this out and we apologize for the negligence. We have carefully checked the numbers presented in the tables and the figures. We have corrected the tables in the revision to resolve the inconsistency and the updated results are presented in Table S2, which now match with the presentation shown in Figure 1 and Figure S2. Nonetheless, our main results are unaffected.

2. Model. This paper conducts linear probability models. However, for dummy dependent variables, is it more appropriate to use Probit or Logit model? Can you explain the reasons for choosing linear probability models and use the results of the Logit and Probit models as robustness checks?

Response: We thank the reviewer for this comment. We agree with the reviewer that a Probit or Logit model would be theoretically more suitable when using dummy dependent variables or for

our desired interpretation (the change of probability in purchasing an Energy Star product). However, when the Probit/Logit model is combined with a series of fixed effects, the fixed effects no longer have desired properties as time-invariant unit level confounders will not cancel out due to nonlinearity of the Probit/Logit model. In addition, when Probit/Logit model is combined with a series of (especially high dimensional) fixed effects, the model may encounter convergence issues due to the number of dependent variables. Therefore we resort to GLS as a compromise similar to the previous studies studying similar topics. We have included more explanations in the third paragraph of the subsection “Empirical strategy” in the section of Methods and as below:

“... energy efficiency. Concerning the choice of model, a Probit or Logit model would be ideal for a dummy dependent variable such as in our case. However, the fixed effects no longer have desired properties as time-invariant unit level confounders will not cancel out due to nonlinearity of the Probit/Logit model. Moreover, rarely do such models converge after including the long list of fixed effects which are also essential in controlling for unobservable against possible confounding effects. Therefore, we estimate the model based on the generalized least square (GLS) method as a compromise following the previous studies^{7,8} using the reghdfe command in Stata. In order to ...”

3. Data. In line 105-107, since the common prices for air conditioners have been listed as \$150-\$750 and deleted records with transaction prices below \$100, why not delete records above a certain dollar?

Response: We thank the reviewer for the comment. We did not delete records above a certain dollar because the maximum price shown in the dataset is \$699.99, below the upper limit indicated above. To address the reviewer’s comment, we have added the below clarification in the third paragraph of the section “Effects of weather on Energy Star Air Conditioner Purchase”,

“... less than \$100 (no records with abnormally large prices are dropped as the maximum price in the dataset is \$699.99 as shown in the descriptive statistics in Table S) and rerun the regression. ...”

We have also included this information in the table of descriptive statistics as suggested in comment 8 below. Please see more details in our response to that point.

4. In Table-S1(3)(4), the author uses the temperature deviation from 21°C as regressors to estimate the marginal effects. How about the nonlinear effects of temperature deviation? Authors could try to use linear and quadratic terms of temperature deviation as regressors. Similarly, incorporating the linear and quadratic terms of meteorological variables into the model can also be used as robustness checks.

Response: We thank the reviewer for this comment. Following the reviewer’s suggestion, we have rerun the regression with the quadratic terms of the temperature deviation and other meteorological indicators included. Results are shown in Table S5 and below. The coefficients of

the temperature deviation are similar; the quadratic terms are insignificant except for the deviation of turning cold in the regressions containing potential price outliers, capturing the firstly increased but then dropped probability of the Energy Star transaction when the temperature lowers down (the turning point happens at around 9.5°C). This trend aligns with Figure 1. We add more interpretations on the additional results in the section “Robustness checks” as below:

“Since temperature deviation as well as meteorological indicators other than the relative humidity may also have a non-linear effect, we introduce the quadratic terms of each weather indicator and rerun the regression. As shown in Table S5, the effect of temperature deviation from 21°C by and large stay linear as in the main analysis, although the probability of Energy Star transactions first increases and then decreases when the temperature drops in the full sample, aligning with Figure 1 that indicates a potential shift to other heating measures when the temperature becomes lower than around 9.5°C. The introduction of the quadratic terms of other meteorological indicators does not change these results.”

Table S5 Robustness check, nonlinear effect of meteorological indicators

	(1) Main	(2) Price Outlier	(3) Main	(4) Price Outlier
(Temperature-21°C)/10	0.050*** (0.013)	0.052*** (0.016)	0.050*** (0.013)	0.052*** (0.016)
(Temperature-21°C) ² /100	-0.005 (0.014)	-0.002 (0.017)	-0.005 (0.014)	-0.002 (0.017)
(21°C-Temperature)/10	0.040*** (0.010)	0.027** (0.011)	0.040*** (0.010)	0.027** (0.012)
(21°C-Temperature) ² /100	-0.021*** (0.007)	-0.009 (0.008)	-0.021*** (0.007)	-0.009 (0.008)
Wind speed	-0.002 (0.002)	-0.002 (0.002)	-0.003 (0.005)	-0.001 (0.005)
Wind speed ²			0.000 (0.001)	-0.000 (0.001)
Precipitation	-0.001*** (0.000)	-0.002*** (0.000)	-0.001** (0.001)	-0.002*** (0.001)
Precipitation ²			0.000 (0.000)	0.000 (0.000)
Relative humidity	-0.005 (0.005)	-0.010 (0.006)	-0.005 (0.005)	-0.010 (0.006)
Relative humidity ²	0.001 (0.000)	0.001*** (0.000)	0.001 (0.000)	0.001*** (0.000)
lnprice	0.097*** (0.005)	0.006 (0.016)	0.097*** (0.005)	0.006 (0.016)
Fixed effects				
County*week	Y	Y	Y	Y
State*year*month	Y	Y	Y	Y

N	1871472	1564506	1871472	1564506
R-sq	0.187	0.196	0.187	0.196

Notes: Standard errors in parentheses are clustered to store level. Results weighted by the units sold per transaction.
* p<0.1, ** p<0.05, *** p<0.01. The relative humidity multiplied by 10.

5. Robustness checks. What is the basis for selecting a comfort zone of 20-22 degrees C and a calculated threshold of 70 degrees Fahrenheit for CDDs and HDDs for this paper? It is recommended to consider robustness checks for different comfort intervals and thresholds and thus investigate whether the results depend on the setting of both. Also, are the results robust to different clustering choices?

Response: We thank the reviewer for this comment. The basis for selecting a comfort zone of 20-22°C is empirical as we tested different intervals as the default group for regression and found that the coefficients of all the other intervals are positive when 20-22°C is adopted. This is close to the comfort zone of 16 to 18°C identified in the previous studies⁹ and the threshold of 65 °F (about 18°C) that is normally used for calculating HDDs and CDDs. We agree with the reviewer adding more tests by moving the reference point would help back up the robustness of our conclusions and have conducted the regression using the 16-18°C as the comfort zone and 65 °F as the threshold for CDDs and HDDs. Additional results are included in Table S3 listed below. We also summarized the results in this table in the section “Robustness checks”:

“As the reference temperature adopted in the analysis (20-22°C for intervals and 70°F for CDDs and HDDs) is slightly higher than what is used in the previous research⁹ (16-18°C for intervals and 65°F for CDDs and HDDs), we switch to these reference interval/threshold to test whether our key findings still hold. The results shown in Table S4 indicate similar statistical significance and magnitudes.”

Table S4 Robustness check, 18°C as reference point

	(1) Main	(2) Price Outlier	(1) Main	(2) Price Outlier
<0°C	-0.007** (0.003)	-0.009** (0.004)		
0-2°C	-0.175*** (0.041)	-0.154*** (0.053)		
2-4°C	-0.029 (0.019)	-0.016 (0.021)		
4-6°C	-0.018 (0.015)	-0.009 (0.016)		
6-8°C	0.007 (0.010)	0.017 (0.011)		
8-10°C	0.023*** (0.009)	0.031*** (0.010)		
10-12°C	-0.003 (0.006)	-0.001 (0.007)		

12-14°C	0.011** (0.004)	0.015*** (0.005)		
14-16°C	0.005 (0.004)	0.008** (0.004)		
18-20°C	-0.001 (0.003)	-0.008** (0.004)		
20-22°C	-0.000 (0.003)	0.000 (0.003)		
22-24°C	-0.008** (0.004)	-0.003 (0.005)		
24-26°C	0.018*** (0.005)	0.019*** (0.006)		
26-28°C	0.018*** (0.006)	0.018*** (0.007)		
28-30°C	0.020*** (0.006)	0.022*** (0.007)		
30-32°C	0.015* (0.009)	0.020* (0.010)		
32-34°C	0.050*** (0.014)	0.063*** (0.018)		
>34°C	0.048** (0.019)	0.063*** (0.021)		
CDD, 65 °F			0.002*** (0.000)	0.003*** (0.001)
HDD, 65 °F			0.002*** (0.000)	0.003*** (0.001)
Wind speed	-0.002 (0.002)	-0.002 (0.002)	-0.002 (0.002)	-0.002 (0.002)
Precipitation	-0.001*** (0.000)	-0.002*** (0.000)	-0.001*** (0.000)	-0.002*** (0.000)
Relative humidity	-0.006 (0.005)	-0.011* (0.006)	-0.008 (0.005)	-0.012* (0.006)
Relative humidity ²	0.001* (0.000)	0.001*** (0.000)	0.001** (0.000)	0.002*** (0.000)
lnprice	0.097*** (0.005)	0.006 (0.016)	0.097*** (0.005)	0.006 (0.016)
Fixed effects				
County*week	Y	Y	Y	Y
State*year*month	Y	Y	Y	Y
N	1871472	1564506	1871472	1564506
R ²	0.187	0.196	0.187	0.196

Notes: Standard errors in parentheses are clustered to store level. Results weighted by the units sold per transaction.
 * p<0.1, ** p<0.05, *** p<0.01. The CDD, HDD, and relative humidity multiplied by 10.

We also thank the reviewer for the suggestion of adding more robustness checks with different clustering levels. We have changed the clustering to county level, the first 3 digits of zip code, and more than 200 Nielsen Designated Market Areas which describe regions in which people get the same options of major media such as television and radio, and are thus faced with the similar advertisement. Additional results are shown in Table S4. We also updated and added explanations to these results in the last paragraph of the section “Robustness checks”:

“In case our results are sensitive to the setting of standard error clustering (store level), we test different clustering levels available in the dataset including county, the first 3 digits of the zip code zone, and the Designated Market Areas (defined by Nielsen to divide the country into more than 200 regions in which consumers are faced with same options of major media and thus the similar information flows from advertisement). The significance of CDDs and HDDs remain with the p-value smaller than 0.001 in all the tests (Table S6).”

Table S6 Robustness check, different levels of clustering

	(1) County	(2) Zip Code, 3 digit	(3) Nielsen Designated Market Areas
CDD	0.004*** (0.001)	0.004*** (0.001)	0.004*** (0.001)
HDD	0.002*** (0.000)	0.002*** (0.000)	0.002*** (0.001)
Wind speed	-0.003 (0.002)	-0.003 (0.002)	-0.003 (0.003)
Precipitation	-0.001*** (0.000)	-0.001*** (0.000)	-0.001*** (0.000)
Relative humidity	-0.005 (0.007)	-0.005 (0.006)	-0.005 (0.011)
Relative humidity ²	0.001 (0.001)	0.001 (0.001)	0.001 (0.001)
Inprice	0.097*** (0.008)	0.097*** (0.008)	0.097*** (0.015)
Fixed effects			
County*week	Y	Y	Y
State*year*month	Y	Y	Y
N	1871472	1871472	1871472
R ²	0.187	0.187	0.187

Notes: Results weighted by the units sold per transaction. * p<0.1, ** p<0.05, *** p<0.01. The CDD, HDD, and relative humidity multiplied by 10.

6. Socioeconomic characteristics. Do socioeconomic characteristics include age and household characteristics, etc.? The impact of demographics on ENERGY STAR air conditioner purchases

may be substantial and bias the estimates if demographics are not randomly assigned among locations.

Response: We thank the reviewer for this comment. We agree that socioeconomic characteristics would affect the type of air conditioners purchased in each transaction. Unfortunately such information is not available in our dataset, so we are not able to include them as control variables. Nevertheless, since we have included the county*week-of-year fixed effect and state*year*month fixed effect, the socioeconomic difference at the county level, and its dynamics at the state level have been controlled and does not bias our estimates.

Nevertheless, we agree with the reviewer that the socioeconomic characteristics may differ the response of consumers to the temperature. This has been addressed in some of our tests on heterogeneous effects. Regarding the suggestion from the reviewer, we have run the same analysis on subgroups divided by median age, median number of rooms, ownership of housing units (the ratio of owner to renter), and the heating fuel of the housing unit (the ratio of electricity to all). The results of heterogeneous effect tests are shown in Figure 3 and Table S10-Table S13, with interpretation of the newly added variables in the second and third paragraph of the section “Heterogenous effects of temperature”:

“... The proportion of white people does not affect the level of response (Column 3, Table S11). Higher age is related to a larger response (Column 3, Table S11), possibly because elder individuals are more sensitive to the temperature or the positive correlation between age and income.

Housing characteristics may differ the response as well. We examined the effect of the median number of rooms of housing units, the ratio of owners to renters, and the percentage of electricity as the heating fuel (Figure 3-h to j). We find that the rooms of housing units do not significantly affect the level of response (Column 1, Table S12), despite that central heating system may benefit more housing units with more rooms and discourage the purchase of window air conditioners. Counties with more owners have a larger response to CDDs but a slightly smaller response to HDDs (Column 2, Table S12), likely implying the presence of principal-agent issues³. A higher proportion of housing units using electricity as heating fuel leads to a smaller response (Column 3, Table S12), indicating the adoption of central temperature control systems. ”

Figure 3 Heterogeneous effect of temperature in terms of CDDs and HDDs. The vertical lines show the 95% confidence intervals. The groups are divided using tertiles of the variable that indicates potential heterogeneous preference.

Table S11 Heterogeneous effect, socio-economic characteristics

	(1) Income	(2) Education	(3) Ethnic	(4) Age
CDD	0.002 (0.001)	0.002* (0.001)	0.004*** (0.001)	0.003** (0.001)
CDD*group 2	0.001 (0.002)	0.001 (0.002)	-0.002 (0.002)	0.000 (0.002)
CDD*group 3	0.003* (0.002)	0.003 (0.002)	0.001 (0.002)	0.003 (0.002)
HDD	-0.001 (0.001)	-0.001 (0.001)	0.003*** (0.001)	0.000 (0.001)
HDD*group 2	0.003*** (0.001)	0.002** (0.001)	-0.001 (0.001)	0.001 (0.001)
HDD*group 3	0.003*** (0.001)	0.005*** (0.001)	-0.003*** (0.001)	0.002 (0.001)
Wind speed	0.002 (0.003)	-0.003 (0.003)	0.005* (0.003)	0.002 (0.003)
Wind speed*group 2	-0.002 (0.004)	0.006* (0.004)	-0.003 (0.004)	-0.001 (0.004)
Wind speed*group 3	-0.002 (0.004)	0.002 (0.004)	-0.013*** (0.004)	-0.004 (0.004)
Precipitation	-0.001 (0.001)	0.000 (0.001)	-0.001 (0.001)	-0.000 (0.001)

Precipitation*group 2	0.001 (0.001)	-0.002* (0.001)	-0.001 (0.001)	-0.001 (0.001)
Precipitation*group 3	-0.002** (0.001)	-0.003** (0.001)	-0.002** (0.001)	-0.003** (0.001)
Relative humidity	0.013 (0.012)	0.002 (0.011)	-0.003 (0.008)	-0.001 (0.008)
Relative humidity*group 2	-0.017 (0.017)	-0.001 (0.017)	0.009 (0.012)	-0.008 (0.013)
Relative humidity*group 3	-0.040** (0.016)	-0.026 (0.017)	-0.028* (0.014)	-0.020 (0.018)
Relative humidity ²	-0.001 (0.001)	0.000 (0.001)	0.001 (0.001)	0.000 (0.001)
Relative humidity ² *group 2	0.002 (0.001)	0.001 (0.001)	-0.001 (0.001)	0.000 (0.001)
Relative humidity ² *group 3	0.003*** (0.001)	0.002 (0.001)	0.002 (0.001)	0.002 (0.001)
lnprice	0.080*** (0.005)	0.080*** (0.005)	0.080*** (0.005)	0.080*** (0.005)
Fixed effects				
County*week	Y	Y	Y	Y
State*year*month	Y	Y	Y	Y
N	1474685	1474685	1474685	1474685
R ²	0.208	0.208	0.208	0.208

Notes: Standard errors in parentheses are clustered to store level. Results weighted by the units sold per transaction.
* p<0.1, ** p<0.05, *** p<0.01. The CDD, HDD, and relative humidity multiplied by 10.

Table S12 Heterogeneous effect, housing characteristics

	(1) Rooms	(2) Owner	(3) Fuel
CDD	0.003** (0.001)	0.003** (0.001)	0.006*** (0.002)
CDD*group 2	0.002 (0.002)	0.000 (0.002)	-0.003* (0.002)
CDD*group 3	0.002 (0.002)	0.004* (0.002)	-0.005*** (0.002)
HDD	0.002* (0.001)	0.003*** (0.001)	0.002** (0.001)
HDD*group 2	-0.001 (0.001)	-0.001 (0.001)	-0.000 (0.001)
HDD*group 3	0.000 (0.001)	-0.003** (0.001)	-0.001 (0.001)
Wind speed	0.001 (0.002)	0.003 (0.003)	-0.001 (0.003)
Wind speed*group 2	-0.002 (0.004)	-0.001 (0.003)	0.004 (0.004)

Wind speed*group 3	-0.004 (0.004)	-0.009** (0.004)	0.000 (0.003)
Precipitation	-0.000 (0.001)	0.000 (0.001)	-0.002*** (0.001)
Precipitation*group 2	-0.003** (0.001)	-0.003*** (0.001)	-0.000 (0.001)
Precipitation*group 3	-0.001 (0.001)	-0.002* (0.001)	0.003*** (0.001)
Relative humidity	0.005 (0.007)	0.011 (0.007)	-0.036*** (0.013)
Relative humidity*group 2	-0.036** (0.015)	-0.027** (0.012)	0.042*** (0.015)
Relative humidity*group 3	-0.042** (0.017)	-0.057*** (0.014)	0.043*** (0.017)
Relative humidity ²	-0.000 (0.001)	-0.001 (0.001)	0.003*** (0.001)
Relative humidity ² *group 2	0.003** (0.001)	0.002** (0.001)	-0.004*** (0.001)
Relative humidity ² *group 3	0.003** (0.001)	0.005*** (0.001)	-0.004*** (0.001)
lnprice	0.081*** (0.005)	0.081*** (0.005)	0.081*** (0.005)
Fixed effects			
County*week	Y	Y	Y
State*year*month	Y	Y	Y
N	1474685	1474685	1474685
R ²	0.208	0.208	0.208

Notes: Standard errors in parentheses are clustered to store level. Results weighted by the units sold per transaction. * p<0.1, ** p<0.05, *** p<0.01. The CDD, HDD, and relative humidity multiplied by 10.

7. In line 106-108, it is not clear on the explanation of how the control of the week's weather indicators addresses the problems that may be caused by the geographic distribution of Energy Star differing from that of non-Energy Star air conditioners. For example, why not control for the geographic location of where air conditioners are purchased, etc.

Response: We thank the reviewer for this comment. The reviewer raises an important question regarding a potential geographic confounder where the geographic distribution of Energy Star products can be different from the non-Energy Star products. If such difference is affected by the weather, the significance testified by our models would be attributed to the accessibility of the Energy Star products in particular stores instead of the psychological mechanisms of the consumers. However, our models effectively control for potential geographical confounders for several reasons. First, the model that we adopt includes the county*week-of-year fixed effect, which controls for the geographic location of where air conditioners are purchased as the reviewer suggested. Therefore, our estimation can be interpreted as a “within effect” that reflects

whether the change of temperature within each county affects the probability of a transaction to be Energy Star or not. Second, while the climate may lead to differential distribution of Energy Star products from the non-Energy Star products in the long run, this effect has been captured in the state*year*month fixed effect, leaving the effect of short-run weather that we are interested. Third, such a difference of geographic distribution may affect the level of response for the consumers living in areas with different background climates, which has been explored in our tests of heterogeneous effects.

To address the reviewer’s concerns, we have added more explanations to further clarify our estimation assumptions in the second paragraph of the subsection “Empirical strategy” in the section of Methods as below:

“... at the county/state level. Specifically, the geographic distribution of Energy Star products may be different from the non-Energy Star products as a result of the weather in the long run. If this holds, the effect of weather can be attributed to the accessibility of the Energy Star products in particular stores instead of the psychological mechanisms of the consumers. The two sets of fixed effects included in the model capture such differences across counties and over time in the long run, and thus enable us to isolate the effect of the short-run weather on consumer decisions. The error term is ...”

8. The summary statistics table is needed.

Response: We thank the reviewer for this comment. Following the reviewer’s comments, we have included the summary statistics in Table S1 as below. We also referred to this table at the end of the sub-section “Empirical strategy” in the methods section:

“... meteorological indicators. The descriptive statistics of the variables adopted in the analysis can be found in Table S1.”

Table S1 Descriptive statistics

	N	Mean	SD	Min	Max
Transaction level					
Price (\$)	2154198	175.44	87.114	0.01	699.99
Energy Star	1608344	170.196	94.762	0.01	699.99
Non- Energy Star	545854	190.892	56.31	0.01	649.99
County level					
Temperature (°C)	1945092	19.841	5.011	-21.036	40.5
CDDs	1945092	19.816	26.007	0	244.3
HDDs	1945092	35.818	45.042	0	531.05
Wind speed (m/s)	1920585	2.914	0.934	0	21.8
Precipitation (mm)	1944655	2.351	3.067	0	67.157
Relative Humidity (%)	1883615	66.267	11.458	0.3	100
Residential electricity price (\$/kwh)	2154197	12.456	2.786	6.88	15.803
CDDs, 1996-2005	2151620	8.753	8.337	0	70.325
HDDs, 1996-2005	2151620	125.552	34.171	9.454	216.324

Median annual household income (\$)	1674221	6.70E+04	1.90E+04	2.00E+04	1.40E+05
Population with bachelors above 25 (%)	1674221	0.338	0.109	0.054	0.753
Population of white (%)	1674221	0.733	0.179	0.155	0.998
Median age (Years)	1674221	39.041	3.776	22.7	59.1
Median number of rooms	1674221	5.557	0.696	3.3	7.7
Owner : Renter (Ratio)	1674221	2.04	0.998	0.236	8.599
Electricity as the heating fuel (%)	1674221	0.234	0.197	0.012	0.975
Population believing climate change happening	2151765	66.258	5.807	43.9	80
Population believing climate change will harm the US (%)	2151765	55.176	4.979	41.6	64.8
Population worried about climate change (%)	2151765	55.597	6.694	37.9	70.3
Population supporting renewable energy standards (%)	2151765	57.659	4.387	40.1	66.2
Population supporting regulation of CO ₂ as a pollutant (%)	2151765	69.246	3.836	49.8	77.9
Support Democrats in election (%)	2151143	0.551	0.158	0.048	0.925

9. Please check the sequence numbers of Figure and Table in the text. For example, in page 6, all "Figure 2" should be revised to "Figure 3".

Response: We thank the reviewer for pointing it out and apologize for the negligence. We have checked all the sequence numbers of the Figures and Tables to ensure that they are referenced correctly.

10. In Equation (1), W_{rt} should be revised to $W_{rt} \delta$.

Response: We thank the reviewer for pointing it out We have added δ in Equation (1) as the coefficient vector of the meteorological variables.

11. In line 126-128, why choose one month before the transaction and three weeks after the transaction, instead of the same period?

Response: We thank the reviewer for this comment. We chose the same periods before and after the transaction. For the lagged weeks, we are adding the weather in 1, 2, and 3 weeks before the period of interest which is the week before the purchase takes place, therefore there are indicators of 4 weeks in total; for the monthly aggregation, we are using the average of these four weeks. We apologize for the confusion and have included more explanation in the second paragraph of the section "Robustness checks":

“... 1) adding the weather indicators in 1, 2, and 3 weeks before the period of interest referring to a previous study (Figure 2-c, Column 5 and 6 in Table S3), and 2) aggregating the weather indicators in the 4 weeks covered in 1), i.e. in the month before a transaction takes place (Figure 2-c, Column 5 and 6 in Table S3). ...”

12. Figure 3. “Bckground climate”, is this a typo here?

Response: We thank the reviewer for pointing out the typo. We have corrected the typo in the updated Figure 3 and proofread the manuscript multiple times to facilitate communications.

- 1 Obradovich, N. & Fowler, J. H. J. N. H. B. Climate change may alter human physical activity patterns. **1**, 1-7 (2017).
- 2 Allcott, H., Wozny, N. J. R. o. E. & Statistics. Gasoline prices, fuel economy, and the energy paradox. **96**, 779-795 (2014).
- 3 Helfand, G. & Wolverton, A. Evaluating the consumer response to fuel economy: A review of the literature. (2009).
- 4 Sallee, J. M. J. T. J. o. L. & Economics. Rational inattention and energy efficiency. **57**, 781-820 (2014).
- 5 Rapson, D. Durable goods and long-run electricity demand: Evidence from air conditioner purchase behavior. *Journal of Environmental Economics and Management* **68**, 141-160, doi:<https://doi.org/10.1016/j.jeem.2014.04.003> (2014).
- 6 U.S. Energy Information Administration. (2005-2020).
- 7 Liao, Y. Weather and the Decision to Go Solar: Evidence on Costly Cancellations. *Journal of the Association of Environmental Resource Economists* **7**, 1-33 (2020).
- 8 Busse, M. R., Pope, D. G., Pope, J. C. & Silva-Risso, J. The psychological effect of weather on car purchases. *The Quarterly Journal of Economics* **130**, 371-414 (2015).
- 9 Albouy, D., Graf, W., Kellogg, R. & Wolff, H. Climate amenities, climate change, and American quality of life. *Journal of the Association of Environmental Resource Economists* **3**, 205-246 (2016).

REVIEWERS' COMMENTS

Reviewer #1 (Remarks to the Author):

The authors completely answered to all my concerns.

Reviewer #2 (Remarks to the Author):

I would like to thank the authors for responding to my comments and amending the paper accordingly.

However, as discussed by the authors, the study is not covering AC penetration explicitly and thus to give a "true" sense of the impact of weather change on final AC energy consumption. To my understanding, the estimates presented in the results section are thus only valid if interpreted as relative contributions.

All in all - despite targeting interesting questions - I believe that the study is falling short of a crucial dimension that would be required to fit into a high-impact journal like Nature Communications.

Reviewer #3 (Remarks to the Author):

This version has been properly revised.

REVIEWER COMMENTS

Reviewer #1 (Remarks to the Author):

The authors completely answered to all my concerns.

We would like to thank the reviewer again for the thorough review that helps us improve the quality of this research.

Reviewer #2 (Remarks to the Author):

I would like to thank the authors for responding to my comments and amending the paper accordingly.

However, as discussed by the authors, the study is not covering AC penetration explicitly and thus to give a "true" sense of the impact of weather change on final AC energy consumption. To my understanding, the estimates presented in the results section are thus only valid if interpreted as relative contributions.

All in all - despite targeting interesting questions - I believe that the study is falling short of a crucial dimension that would be required to fit into a high-impact journal like Nature Communications.

We thank the reviewer for this comment. We agree with the reviewer that our estimates are only valid if interpreted as relative understanding and we are not able to cover the AC penetration explicitly. In this way, we provide the back-of-envelope calculation (1°C increase and decrease from 21°C would help to save electricity use by 292.4 and 146.2 billion Btu) as the best we can do to demonstrate the significance of our findings at the macro-level, which is not trivial. Moreover, the main contribution of this study is that it provides empirical evidence on whether and how short-term change of temperature could help overcome the energy efficiency gap as a “low-hanging fruit” to facilitate policy making and marketing strategies of air conditioning in a more cost effective way, which is underexplored in the current literature. With our estimates, the nationwide impact of weather change can be definitely calculated with improved accuracy in the future studies if the penetration data is accessible.

Reviewer #3 (Remarks to the Author):

This version has been properly revised.

We would like to thank the reviewer again for the valuable comments that helps us improve the quality of this research.